# In Vitro Anti-Venom Potentials of Aqueous Extract and Oils of *Toona ciliata* M. Roem against Cobra Venom and Chemical Constituents of Oils

**DOI:** 10.3390/molecules28073089

**Published:** 2023-03-30

**Authors:** David Fred Okot, Jane Namukobe, Patrick Vudriko, Godwin Anywar, Matthias Heydenreich, Oyedeji Adebola Omowumi, Robert Byamukama

**Affiliations:** 1Department of Chemistry, Makerere University, Kampala P.O. Box 7062, Uganda; 2Centre for Snakebites and Venom Research, Department of Chemistry, Gulu University, Gulu P.O. Box 166, Uganda; 3Research Centre for Tropical Diseases and Vector Control, Department of Veterinary Pharmacy, Clinics and Comparative Medicine, College of Veterinary Medicine, Animal Resources and Biosecurity, Makerere University, Kampala P.O. Box 7062, Uganda; 4Department of Plant Sciences, Microbiology & Biotechnology, Makerere University, Kampala P.O. Box 7062, Uganda; 5Institute of Chemistry, University of Potsdam, Karl-Liebknecht-Str. 24-25, D-14476 Potsdam, Germany; 6Department of Chemical & Physical Sciences, Faculty of Natural Sciences, Walter Sisulu University, Private Bag X1, Mthatha 5099, South Africa

**Keywords:** snakebite, venom, medicinal plants, anticoagulation, antiphospholipase a_2_ inhibitor, rutamarin and seselin

## Abstract

There are high mortality and morbidity rates from poisonous snakebites globally. Many medicinal plants are locally used for snakebite treatment in Uganda. This study aimed to determine the in vitro anti-venom activities of aqueous extract and oils of *Toona ciliata* against *Naja melanoleuca* venom. A mixture of venom and extract was administered intramuscularly in rats. Anticoagulant, antiphospholipase A_2_ (PLA_2_) inhibition assay, and gel electrophoresis for anti-venom activities of oils were done. The chemical constituents of the oils of *ciliata* were identified using Gas chromatography-tandem mass spectroscopy (GC-MS/MS). The LD_50_ of the venom was 0.168 ± 0.21 µg/g. The venom and aqueous extract mixture (1.25 µg/g and 3.5 mg/g) did not cause any rat mortality, while the control with venom only (1.25 µg/g) caused death in 1 h. The aqueous extract of *T. ciliata* inhibited the anticoagulation activity of *N. melanoleuca* venom from 18.58 min. to 4.83 min and reduced the hemolytic halo diameter from 24 to 22 mm. SDS-PAGE gel electrophoresis showed that oils completely cleared venom proteins. GC-MS/MS analysis showed that the oils had sesquiterpene hydrocarbons (60%) in the volatile oil (VO) and oxygenated sesquiterpenes (48.89%) in the non-volatile oils (NVO). Some major compounds reported for the first time in *T. ciliata* NVOs were: Rutamarin (52.55%), β-Himachalol (9.53%), Girinimbine (6.68%) and Oprea1 (6.24%). Most compounds in the VO were reported for the first time in *T. ciliata,* including the major ones Santalene (8.55%) and Himachal-7-ol (6.69%). The result showed that aqueous extract and oils of *T. ciliata* have anti-venom/procoagulant activities and completely neutralized the venom. We recommend a study on isolation and testing the pure compounds against the same venom.

## 1. Introduction

The World Health Organization classified snakebite envenomation as one of the most neglected tropical diseases (NTD) in terms of incidence, severity, and clinical characteristics [1]. This has served as a basis for its advocacy and enlisting as an NTD under category A [1,2]. In Sub-Saharan Africa, Snakebite envenoming constitutes a serious medical condition that primarily affects residents of rural communities including Asia, Latin America, and New Guinea Or we use Snake bite is a common and frequently devastating environmental and occupational disease, especially in rural areas of tropical developing countries [3]. It is an occupational, environmental, and domestic health hazard that exacerbates the already impoverished state of these communities [4]. Since the comprehensive review by Swaroop and Grab in 1954, no global survey has been carried out on snakebite epidemiology. The present article of 2012 was an attempt to draw the attention of health authorities to snake envenomations and urges them to prepare therapeutic protocols adapted to their needs [5]. Available statistical data are known to be unreliable and at best can serve to provide only an approximate and highly conservative estimate of the relative magnitude of the snakebite problem [6].

Conservative estimates indicate that, worldwide, more than five million people suffer snakebite every year, leading to 25,000–125,000 deaths, while an estimated 400,000 people are left with permanent disabilities [4,7]. The recorded incidence of snake bites in Bouenza was 18.62 per 100,000 of the population of the Republic of Congo [8]. In Rwanda, it is uncertain how many people are envenomed by snakes each year. One model, based on regional data, estimated that 4171 cases (95%) and 196 deaths (95%) occur throughout the country each year, according to the work of Halilu et al. (2019) [9]. However, in Rwanda, it was reported that by 2017 and 2018, only 182 Snake bites envenomations cases were annually reported by hospitals [10].

According to a newspaper report in Uganda, it is believed that 30 000 people die from snakebites yearly in Africa [11]. New Vision Uganda newspaper reported that snakebites per year in Kamuli included 349 cases, Mubende had 326 cases, Yumbe had 395 cases, and Amuru had 283 cases of Snakebites [12].

Venoms are mainly toxic modified saliva of all poisonous snakes consisting of a complex mixture of enzymes, protein, non-protein, and metalloproteinase [13]. The most important venom components that cause serious clinical effects are procoagulant enzymes, cytolytic or necrotic toxins, hemolytic and myolysis phospholipases A_2_, pre- and post-synaptic neurotoxins, and hemorrhages [14]. Snake venom inhibits the coagulation of blood, resulting in profound bleeding through the wound of the bitten animal [15]. Venom proteins with enzymatic activity, such as phospholipases A_2_ and proteinases, inhibit blood coagulation [15]. Phospholipases A_2_ (PLA_2_s) are commonly found in snake venoms from viperidae, hydrophidae, and elaphidae families (e.g., *Naja melanoleuca*). The PLA_2_s have been extensively studied due to their pharmacological and physiopathological effects, such as pre or post-synaptic neurotoxicity, cardiotoxicity, myotoxicity, platelet aggregation inhibition, edema, hemolysis, and anticoagulation in living organism [16]. The pathology in elapid snakebite victims derives from neurotoxins, which can cause paralysis of the diaphragm and eventually asphyxiation; that is why this study used *Naja melanoleuca*.

Snake anti-venom immunoglobulins (serum therapy) are the only specific treatment for snakebites. A major drawback of serum therapy is its prohibitive cost and the chance that victims are often some distance away from medical care when bitten [17]. That is why the local community looks for alternatives to medicinal plants.

Other medicinal plants used as anti-cobra venom: Different studies have been performed on Pakistan and Indian medicinal plants against cobra snakebites, but the cobra of species *Naja Naja* is the most common [18,19]. Other studies showed that the ethanolic bark extract of *Buchanania lanzan* significantly neutralized venom *Naja naja* cobra venom with about 50% of hemolysis [20]. In another study, the aqueous seed extracts of *Mucuna pruriens* completely neutralized the venom of *Naja Naja* and krait venom in India [21]. In Pakistan, Citrullus colocynthis, Rubia cordifolia, and Stenolobium stans decreased the anticoagulant of phospholipase from 92 to 20%. Other plant species with mild anticoagulant activities were; Albizia lebbeck, Brassica nigra, Matthiloa incana, Neriunt indicant, and Rhazya stricta [18]. In this study, forest Cobra (Naja melanoleuca) venom was used because of the most reported cases in health centers during the survey.

*Toona ciliata* M. Roem belongs to the Meliaceae family and is commonly known as Toonee, Tuni in Hindi; Red cedar in English; and Nandi in Sanskrit. It is mainly distributed in the tropical Himalayas from the Indus Eastward and throughout the hills of Central and Southern India (Ayurvedic) by Negi 1993 [22]. *Toona cilliata* originates from tropical Asia and tropical Australia but is now much cultivated throughout the tropics for its timber and as an ornamental or as a wayside tree. It was recorded from Zambia and Zimbabwe as early as the beginning of 20 century. It has locally become naturalized in Southern Africa by Loupee D 2008 [22]. The evolutionary divergence between two *Toona* species has been studied, and the result showed that the *T. ciliata* genome was highly collinear with the T. sinensis genome but had low collinearity with the genomes of more distant species [23]. *Toona Cilliata* is traditionally used in Southern Africa by herbalists to treat venereal diseases and has been found to treat cervical cancer, with evidence of induced apoptotic activity in the HeLa cells line when using extracts from *Toona ciliata* and *Seriphium plumosum* [22]. Methanol extracts of *Toona ciliata* exhibited phytopathogenic bacteria and significant antifungal activity against *Microsporum canis* [24]. Another use of *T ciliata* other than medicinal was in biodiesel, and membrane technology was employed to synthesize biodiesel from *Toona ciliate* novel and non-edible seed oil. The optimal yield of biodiesel attained was 94% at 90 °C for 150 min [25].

The essential oil of *T. ciliata* Roem. has been found to be effective as an antidepressant which can be used as a potential antidepressive drug in the clinic in the future [26]. Essential oils were previously identified in the leave (L), and the stem *of Toona ciliata* was detected but not in the roots bark of *Toona ciliata*. They include; Sesquiterpenes: Cubebene (L), Ylangene (L), Cubebene (L), β-Elemene (L&St), α-Guaiene (L), and Oxygenated sesquiterpenes: Cubebol (L), Globulol (L&St), epi-Cubenol (L), Muurolol (L) and Cadinol (L&St) [26,27].

The root bark of *T. ciliata* is used for the treatment of many health conditions, including snakebites in Uganda [28]. *T. ciliata* has been used for the treatment of snakebites in Northern Uganda for a decade. The local name in the Luo language is “Yat bwoc or ‘’Yat luu pa coo”, meaning it was used to treat erectile dysfunction. Because they have been treating erectile disfunction with its root bark, to date, the species are used by Traditional Medicine Practitioners (TMPs), and the frequency of mention was four out of four (there were only four TMPs in the region), using for the treatment of snakebites. Another local use is for the treatment of erectile dysfunction before trying for snakebites, which provoked the study hence selection [28].

However, to the best of our knowledge, there has been no anti-venom potential and phytochemical screening of *T. ciliata* despite its wide use in traditional medicine for snake envenomation by oral administration of aqueous extract of the root barks. This study aimed to determine anti-venom, anticoagulant, and antiphospholipase A_2_ inhibition activities of the aqueous root bark extracts, carry out in vitro Sodium dodecyl sulfate (SDS) polyacrylamide gel electrophoresis (PAGE) (SDS-PAGE) analysis of non-volatile/volatile oils of *T. ciliata* and carry out chemical constituents profiling of oils from the root bark of *T. ciliata* by GC-MS/MS.

## 2. Results

### 2.1. Phytochemical Screening

Phytochemical screening of both aqueous and organic extracts of root barks of *T. ciliata* was performed, and the results showed that it contained various secondary metabolites, including alkaloids, tannins, saponins, phlorotannins, flavonoids, coumarins, anthraquinones, terpenoids, and glycosides. A detailed description of the phytochemicals has been published as a Mendeley data set vol. 1, 2021.

### 2.2. Lethal Dose (LD_50_) of Venom

Mortality of the rats with their calculated mean survival time at different lethal doses was recorded starting with the higher dose. The dose doses that killed rats in a few seconds were left out (not good for in vivo experimental tests). In this study, the results showed that; A dose of 5 µg/g killed rats with a mean survival time of 0.55 h (h), while a dose of 2.5 µg/g had a mean survival time of 1.0 h. A dose of 1.25 µg/g killed at an increased mean survival time of 2 h. Consequently, as the lethal dose concentrations reduced, the survival times increased until a lethal dose of 0.005 µg/g, where the rats survived for 24 h (Table 1).

#### Probit Analysis to Determine LD_50_

Probit analysis is a specialized form of regression analysis, which is applied to binomial response variables, i.e., variables with only one of two possible outcomes (positive/negative), that is, death/survival in this study.

Probit statistic was used to analyze the data in Table 1, and the result of the calculation is in Table 2.

The result of Table 2 was plotted and graphed (Figure 1). The regression equation was y = 3.3203x + 7.5693 which was used to calculate the LD_50_ = 0.16833886 ± 0.210835 µg/g.

Generally, the result of this study showed that *N. melanoleuca* venom is very lethal since it has LD_50_ as low as 0.16833886 µg/g.

### 2.3. Administration of a Mixture of Venom and Plant Extracts after the Pre-Incubation Test

Dose of control with a concentration of 1.25 mg/kg of venom killed rats with a mean survival time of 2.0 h. The same concentration of venom (1.25 mg/kg) mixed with 3.5 mg/g of aqueous extract of *T. ciliata*, there was no death of rats registered. For the mixture of venom and a higher concentration of extract (5.0 mg/kg), the result showed no mortality after 48 h (Table 3), and the rats were taken to their cages to live normally under observation for more than 4 days.

### 2.4. In vitro Inhibition of Anticoagulant Activities of Venom

#### 2.4.1. Minimum Coagulation Dose of Plasma (MCD-P)

Experiments were performed in triplicate, and the MCD-P dose of 12.5 μg/mL gave a mean clotting time of 5.58 ± 0.44 min, but there were no clots formed for concentrations starting at 40 μg/mL and above. The control (Phosphate buffered saline, PBS) caused coagulation at 3.92 ± 0.38 min (Table 4).

Generally, several proteins with enzymatic activity, such as PLA_2_ and proteinases, inhibit blood coagulation, and this study showed that at high concentrations of *N. melanoleuca* venom ≥40 µg/g, there was inhibition of blood clots.

The effects of extracts on coagulation time revealed that concentrations of 25.5 and 100 mg/mL caused plasma coagulation without recalcification. Concentrations of 12.5, 6.25, 1, 0.5, 0.25, and 0.125 mg/mL did not cause coagulation. However, clots occurred in shorter periods than control samples when recalcified with 25 mM CaCl_2_ (Table 5).

These results point out that higher doses cause coagulation without recalcification, which is unique, but the study required recalcification, which is why the low doses were chosen.

#### 2.4.2. Inhibition of the Anticoagulation Activity of Venom

Neutralization of the anticoagulation activity of venom by the aqueous extract was compared with SAIMR (South African Institute for Medical Research) polyvenom (South African antisera) purchased from joint medical stores in Kampala, Uganda. MCD-P was multiplied by two (2xMCD-P) and constantly used without varying 2xMCD-P (25 μg/mL) caused coagulation for 11 min, while when 2xMCD-P mixed with 6.25, 0.5, and 0.125 mg/mL coagulation occurred at meant clotting time of 22.29, 14.65, and 14.83 and antisera at 8.67 min (Table 6). These indicate that the crude aqueous extract inhibited the coagulation activity of *N. melanoleuca* venom close to antisera as positive control. This is because the extract has proven to be a procoagulant (a precursor of a natural substance necessary for the coagulation of blood).

The results of two experiments showed that a low concentration of *T. ciliata* extract has some activities, but due to using 2xMCD-P from other methods, we recommend using high doses of venom, which inhibits blood clots. In conclusion, some concentrations of aqueous plant extracts cause plasma clots, just as some low concentrations of venom.

### 2.5. Inhibition of Phospholipase A2 (PLA_2_) Activity of Venom

#### 2.5.1. Determination of Minimum PLA_2_ Hemolytic Dose of Venom

In the phospholipase A2 activity assay, *N. melanoleuca* venom was able to produce hemolytic haloes in agarose-human erythrocytes gels. The minimum hemolytic dose (MHD) is defined as the amount of venom that induces a hemolytic halo of 22 mm diameter [15]. *N. melanoleuca* venom had the lowest MHD of 2 μg and 10 μg at 10 mm and 22 mm diameter hemolytic halos, respectively. The MHD of *N. melanoleuca* venom was 10 μg. This shows that *N. melanoleuca* venoms have phospholipase A2 enzymes that can lyse human erythrocytes. The control 30 μL of PBS, pH 7.4, produced 6 mm halos (Table 7).

Hemolytic properties of the aqueous extract of *T. ciliata* root bark on human erythrocytes showed that concentrations of ≤0.5 mg/mL produced no hemolytic halos, but concentrations of 1.0, 6.25, and 12.5 mg/mL produced hemolytic halos of 8, 16 and 18 mm (Table 8).

#### 2.5.2. Inhibition of PLA_2_ Hemolytic Activities of Venom by Extract

10 μg of MHD was mixed with various concentrations (6.25, 1.0, 0.5, 0.25 mg/mL) of the aqueous extract of *T. ciliata*, and the various mixtures were incubated for 30 min at 37 °C. The results showed that concentrations of 6.25, 1.0, 0.25, and 0.5 mg/mL induced hemolytic haloes of 22 mm and 23 mm diameter. The control of 10 μg venom only induced a hemolytic halo of 24 mm diameter (Table 9) and Figure 2A,B.

Hemolytic activities of MHD were neutralized by an aqueous extract with a 2 mm diameter.

Extract at a concentration of ≥1.0 mg/mL causes small hemolytic activities but reduces the hemolytic activity of control (10 μg) from 24 to 22 halos.

### 2.6. Anti-Venom Activities of Non-Volatile and Volatile Oils of T Ciliata by SDS-PAGE

Sodium dodecyl sulfate (SDS) polyacrylamide gel electrophoresis (PAGE) (SDS-PAGE); SDS-PAGE was used to separate proteins with relative molecular masses from venom and protein band observed. Venom mixed with oils; lane three (V + VO) and lane four (V + NVO) there were no protein bands, while lane one Protein marker (PM) and lane two venoms showed protein bands as control (Figure 3).

The good Protein marker (PageRuler™ Plus Prestained Protein Ladder, 10 to 250 kDa) had a good range of KiloDalton (KDa) from 10 to 250, and the equipment was well calibrated, but some of the venom proteins were not very visible, but the freshly milked venom gave the bands hence more research because the Snakes were caged. Generally, when the protein bands are not present in the gel wells after running a mixture of proteins with another extract, it means the extract has neutralized the protein.

### 2.7. Chemical Composition of Non-Volatile and Volatile Oils by GC-MS/MS

The compounds of the oils were identified by matching their spectra and retention indices (Kovats Index) with those of the authentic samples and literature value from the Wiley Library of Mass Spectra database of the GC-MS/MS system and published data.

#### 2.7.1. Phytocompounds in Isolated Non-Volatile Oils

During the isolation of the Organic extract (Dichloromethane: Ethylacetate = 1:1), the non-volatile oils were eluted with a solvent system of (Hexane: Ethylacetate = 4:1 and 3:2), and each of the oils was analyzed for chemical composition by Gas chromatography (GC) coupled with double detector mass spectroscopy (MS/MS) (GC-MS/MS). A total of **24** compounds were analyzed and identified from non-volatile oil **A** (Table 10). The major compounds in A include; Rutamarin (44.34%) and Himachalol (20%), indicated in spectra (Figure 4), with the highest percentage of the rest are Di-n-2-propylpentylphthalate (Phthalic acid, di(2-propylpentyl) ester) (2.77%), Humulene epoxide II (2.67%), 2-Butanone,4-(2,4,6-trimethoxyphenyl) (2.37%), Panasinsene (2.06%), [5-(Hydroxymethyl)-2,5,8a-trimethyl-1,4,4a,6,7,8-hexahydronaphthalen-1-yl]methanol (1.86%). The rest of the compounds with percentages less than 1.86% are listed in Table 10.

A total of 21 compounds were identified (Table 11) in second eluted non-volatile oil **B**, with about 10 major compounds, including Rutamarin (52.55%) and Himachalol (7.61%) indicated in spectra (Figure 5); the others are Guaiene (5.9%), Girinimbine (3.94%), 2,6-Dihydroxybenzoic acid, 3TMS derivative (3.46%), Allohimachalol (3.13%), 9H-Carbazole, 9-methyl- (2.2%), 4,9-Muuroladien-15-ol (1.31%), 1-Aromadendren-10-ol (1.24%), and cis-Z-α-Bisabolene epoxide (1.16%).

The last eluted non-volatile oil **C** had **38** compounds analyzed and identified. The following are the major compounds; Rutamarin (15.15%), Guaiene (10.06%), 4,9-Muuroladien-15-ol (9.53%), Girinimbine (6.68%), Oprea1 (6.24%) (Harmala alkaloids class), Citronellol, TMS derivative (6.18%), 7-Tetradecenal, (Z)- (5.83%), 3-Methylcarbazole (4.21%), Isolapachol (3.98%), and Cadinol (3.61%) (Table 12). The spectra of GC-MS/MS are presented (Figure 6).

Compounds were identified and reported for the first time in non-volatile oils (NVO) of the genus Toona.

In this study, the following are major compounds (16) reported for the first time in the genus Toona in NVOs; Rutamarin (52.55%), Himachalol (7.61%), 4,9-Muuroladien-15-ol (9.53%), Girinimbine (6.68%), Oprea1 (6.24%) (Harmala alkaloids class), Citronellol, TMS derivative (6.18%), 7-Tetradecenal, (Z)- (5.83%), 2,2,4-Trimethyl-3-(3,8,12,16-tetramethyl-heptadeca-3,7,11,15-tetraenyl)-cyclohexanol (4.3%), 3-Methylcarbazole (4.21%), Isolapachol (3.98%), Cadinol (3.61%), Allohimachalol (3.13%), 6-Isopropenyl-4,8a-dimethyl-1,2,3,5,6,7,8,8a-octahydro-naphthalen-2-ol (2.92%), 2-Butanone,4-(2,4,6-trimethoxyphenyl) (2.37%), n-Hexadecanoic acid (2.37%) and Panasinsene (2.06%). Rutamarin (52.55%) is the major compound found in NVOs, and its MS/MS fragments were analyzed (Figure 7).

Figure 8 shows the structure of 16 major compounds, including Rutamarin, which was detected in all the non-volatile oils.

#### 2.7.2. Phytocompounds in the Volatile Oil

Volatile or essential oil (VO) of the root bark of *Toona ciliata* was analyzed, and a total of 49 compounds were identified and reported for the first time in the roots (Table 13, Figure 9). The following are some of the major compounds according to their percentages; Santalene (8.55%), copaene (8.1%), (+)-2-Himachalen-7-ol (6.69%), Longicyclene (5.05%), Longipinene (4.63%), Guaiene (4.44%), Cadina-1(10),4-Diene (4.1%), Copaene (4.68%), 1.Xi.,6.Xi.,7.Xi.-Cadina-4,9-Diene.α.-Amorphene (3.24%), (-)-Trans-Caryophyllene (2.87%), Dihydroagarofuran (2.62%), Cubenol (2.58%), 3,11-Acoradiene (2.21%), DI-Neoisolongifolene (2.21%), Ylangenol (2.12%) and Cyclosativene (2.04%)(>2.0%).

The structure of some of the major compounds, according to their percentages identified in volatile oil, are listed below Santalene (8.55%), Himachalol (6.69%), Longicyclene (5.05%), Cadina-1(10),4-Diene (4.1%), Amorphene (3.24%), Dihydroagarofuran (2.62%), Cubenol (2.58%), 3,11-Acoradiene (2.21%), Neoisolongifolene (2.21%), Ylangenol (2.12%), [5-(Hydroxymethyl)-2,5,8a-trimethyl-1,4,4a,6,7,8-hexahydronaphthalen-1-yl]methanol 2.12%, Cyclosativene (2.04%), 1 (-)-Isolongifolol, acetate(1. 88%), β-Calacorene (1.46%), 4,9-Muuroladien-15-ol (1.2%), 5-Metoxy seselin, and Rutamarin (0.41%) (Figure 10). Most of the compounds presented in this study for the volatile oil of root bark were reported for the first time according to previous research on genus *Toona* [21,22].

During analysis of all the classes of compounds in NVOs and VO, the classes of chemical compounds were Sesquiterpene hydrocarbon, oxygenated sesquiterpene, furanocoumarins, and hydrocarbon alcohol, all of which are present in both NVO and VO (Table 14).

## 3. Discussion

Research to develop a treatment for local envenoming is a clinical priority and has focused on the application of natural products in Uganda. Herbalists have been using plants to make a decoction from either aerial, stems, or roots bark and administered orally in most cases [28,29]. The quantity of the decoction administered is not specific.

### 3.1. Phytochemical Screening

Most of the phytochemicals were detected both in the aqueous and organic extract of *T. ciliata*. Flavonoids and polyphenolic compounds detected in this study have the ability to bind to macromolecules, and some of these have shown the potential to inhibit PLA_2_s in other studies [30]. These include; quercetin, luteolin, kaempferol, isoquercitin and rutin [30]. Pure quercetin-3-O-rhamnoside from *Euphorbia hirta* has been found to have anti-venom activity [31].

In a review of ethnomedicinal plants as a source of phytochemical compounds against snake venom PLA_2_s activity, the alkaloid 12-methoxy-4-methylvoachalotine, the coumarins 7-methoxy-coumarin and 6,7-methylenedioxy-coumarin, the terpenoids betulin and betulinic acid and other tannins inhibit the activities of PLA_2_ venom [30,32].

### 3.2. In Vitro Anti-Venom Potentials of Aqueous Extract of T. ciliata

When aqueous extracts were mixed with a lethal dose of venom, it completely neutralized venom because i.m. administration had no mortality registered. The many compounds could have affected the structure proteins, non-proteins, and enzymes of *N. melanoleuca,* yet there was no effect of solvents. The most life-threatening pathology in elapid snakebite victims derives from neurotoxins, which can cause paralysis of the diaphragm and, eventually, asphyxiation [33]. There is an indication that some pure compounds may have anti-venom potentials if isolated and tested according to the reseach by Kankara et al. (2020). Similar studies were performed in other countries, showing similar in vitro anti-venom potentials of medicinal plants. Some medicinal plants in India inhibited the lethality of *Naja naja* and *Daboia russelii* snake venoms. The results of their study showed snake venom-antagonizing properties of the aqueous extracts of five plants, *viz. Sapindus laurifolius*, *Spondias pinnata*, *Plumeria lutea*, *Woodfordia fruticosa*, and *Croton roxburghii*, with *Woodfordia fruticosa* showed good anti-venom potential consistently in all experiments (in vitro and in vivo) while other are good when mixed with venom and incubated (in vitro) [34].

*Annona senegalensis* Pers (family: Annonaceae) is used traditionally in Nigeria to treat victims of snakebite; it has been found that the extract mixed with venom inhibited (in vitro) the toxic signs induced by the cobra (*Naja nigricotlis nigricotlis* Wetch) venom [35]. The extract of the leaves of *Guiera senegalensis* was found to detoxify (in vitro) venom from two common northern Nigerian snake species, *Echis carinatus* and *Naja nigricollis,* and survival percentage was recorded within 24 h [36].

### 3.3. Inhibition of Anticoagulant Activities of Venom

Bites from elapid snakes typically result in neurotoxic symptoms in snakebite victims. Neurotoxins are, therefore, often the focus of research relating to understanding the pathogenesis of elapid bites. However, recent evidence suggests that some elapid snake venoms contain anticoagulant toxins, which may help neurotoxic components spread more rapidly [37]. African spitting Naja species significantly inhibited thrombin which has the ability to clot human fibrinogen and the impediment of Factor Xa’s ability to clot recalcified plasma by forming a prothrombinase complex with Factor Va [38]. Factor Xa is the trypsin-like proteinase (serine protease enzyme) of coagulation that catalyzes prothrombin activation (prothrombin is a protein known as clotting (coagulation) factors (causes blood to clot) it is made by the liver while thrombin prevents blood clot. Factor Va is an essential protein cofactor of the enzyme factor Xa which activates prothrombin to thrombin during blood coagulation [39]. Venom effects on the coagulation cascade could cause anticoagulant effects, including inhibition of the blood coagulation cascade enzymes thrombin and Factor Xa [9]. Several cobra species have been shown to produce anticoagulant effects by inhibiting blood coagulation factors through the use of modified Group I phosopholipase A_2_ (PLA_2_) toxins [40]. For example, inhibitors of the enzymatic activities of Factor Xa and thrombin have been isolated from the non-spitting African cobra *N. haje* [41].

Many snake venoms comprise of different factors, which can either promote or inhibit the blood coagulation pathway. Coagulation disorders and hemorrhage belong to the most prominent features of bites of the many vipers; fractions of Iranian *Echis carinatus* venom delayed the prothrombine time and thus can be considered anticoagulant factors [42]. Some snake venoms contain toxins that are direct or indirect anticoagulants that inhibit the clotting process, thus increasing the risk of bleeding. However, other species with anticoagulant toxins coexist with coagulant and hemorrhagic toxins, thus producing a far less clear or diagnostic clinical laboratory picture [43].

In this study, it has been found that a high concentration of forest cobra (*N. melanoleuca*) venom from 40 μg/mL and above prevent blood plasma from clotting while dosing from 30 μg/mL and below cause blood plasma clot. The aqueous of *T. ciliata* was made to prolong the coagulant activity of 2xMCD-P (25 μg/mL).

Higher concentrations of the aqueous extract of *T. ciliata* cause blood plasma clots without recalcification, thus proving to be procoagulant (a precursor of a natural substance necessary to coagulate the blood). This could be useful for drug development. The procoagulant activity of *T. ciliata* root bark extract has proven its neutralization potentials that inhibit the anticoagulant activity of *N. melanoleuca* venom through its crude extract with groups of chemical compounds working synergistically. Concentrations (12.5–0.125 mg/mL) do not cause coagulation and are used for experiments with recalcification.

### 3.4. Inhibition of Phospholipase A_2_ Activities

Phospholipase A_2_ (PLA_2_) is a ubiquitous enzyme that is an important component of snake venoms. These enzymes hydrolyze glycerophospholipids at the sn-2 position of the glycerol backbone, liberating lysophospholipids and fatty acids. Snake venom protein PLA_2_s displays a great variety of biological activities, including neurotoxic, myotoxic, antiplatelet, hemorrhagic, and anticoagulant effects [43].

The membranes of the erythrocytes can be affected by the consumption of bioactive compounds from herbs and medicinal plants [44]. In this study, the aqueous extract of *T. ciliata* root bark caused hemolysis of human erythrocytes at concentrations 6.25 and 12.5 mg/mL. This concentration reduced the minimum hemolytic dose (MDH) of 10 μg of PLA_2_ by 8.33%, implying that the plant has an inhibitory effect on PLA_2_ of the forest cobra of Uganda. This result reveals that hemolytic activity was inhibited by 2.0- and 1.0-mm halos in a concentration-dependent manner which implies that the aqueous extract has anti-phospholipase A_2_ hemolytic activities, indicating that hemolysis was reduced by the plant. These results are in agreement with other similar research performed elsewhere. For instance, chloroform extract of *Cyphostemma adenocoule* through in vitro inhibited phospholipaseA_2_ enzyme [45]. Galic acid (GA) and other tannins other isolated from Brazilian *Anacardium humile* have been shown to be effective inhibitors of snake venoms’ toxic effects, and herein we demonstrated GA’s ability to bind to and inhibit a snake venom PLA_2_, thus proposing a new mechanism of PLA_2_ inhibition, and presenting more evidence of GA’s potential as an anti-venom compound [46].

### 3.5. In Vitro Anti-Venom Potentials of Oils of T. ciliata by SDS-PADE

The interest of using SDS-PAGE was to determine the decreased intensity or disappearance of bands as well as the appearance of bands of different molecular weights in the lanes loaded with venom incubated with both oils for reliable indicators of the anti-venom activity or not.

SDS-PAGE was carried out using Protein marker (PM) lane one and venom lane two while the mixture of venom and oils, lane three and lane four, where venom was mixed with volatile and non-volatile oils, then visualized according to the protocol of Laemmli 1970. There were no protein bands seen in both lanes three and four. These indicated that there were no proteins, thus implying that both oils neutralized the *N. melanoleuca* venom proteins. It can be concluded that both oils have anti-venom potential. Volatile oils have a high yield during hydro-distillation and thus can be recommended for anti-*N. melanoleuca* venom, but more research should be performed on other species of Snake venom. Similar research was performed on essential oil and aqueous extract of *Aloysia citriodora* against *Bothrops diporus* venom. The SDS-PAGE results for the inhibition of proteolytic activity showed that essential oil was found to have much more active than plant extracts, but both neutralize the proteolytic activity [47]. *Bothrops atrox* venom proteins were investigated using SDS-PAGE electrophoresis if the proteins formed molecular complexes or were precipitated after being exposed to the *Urospatha sagittifolia*. The result showed that the protein band loses intensity as the dose of the extract is increased. This suggests both the precipitation of proteins and/or the formation of molecular complexes, which can be related to the inhibition of enzymatic activity. Not only SDS-PAGE analysis, Ethanolic extract of *Urospatha sagittifolia* (Araceae) reduces paw edema, skin hemorrhage and lethality induced by the crude venom on mice [48]. In this study, there was complete disappearance of the protein band, which concludes that the oils are potential anti-venom, more especially essential (volatile) oil, which is easily extracted from the root bark of *T. ciliata*.

### 3.6. Chemical Constituents of Non-Volatile and Volatile Oils of T. ciliata Root Bark by GC-MS/MS

The amount of oxygenated sesquiterpene is higher (48.89%) in non-volatile oils (NVOs) than in volatile oil (VO) (25%) due to the fact that they were extracted with n-hexane and concentrated using rotary evaporations and eluted through column chromatography. Volatile oils (VO) underwent only hydro-distillation, and Sesquiterpene hydrocarbon (60%) was the highest.

In NVOs, 83 compounds, out of which 15 major compounds selected by their percentage abundance, are reported for the first time in the genus *Toona* (Meliaceae) in this study about the root bark. However, according to literature, essential oils were previously identified in the leaves and stems of species *Toona ciliata, Cedrela odorata*, and *Cedrela fissilis* of the family Meliaceae [27]. The following compounds are the major ones in NVOs root bark with their medicinal values in relation to anti-venom; Rutamarin (52.55%) was found to be an effective inhibitor of human monoamine oxidase B (*h*MAO-B) with an inhibition percentage of 95.26% and also a study on molecular docking of (S)- rutamarin with *h*MAO- B showed that it binds stronger to the *h*MAO-B binding cavity [49]. Monoamine Oxidase B (MAOB) is involved in the breakdown of dopamine, a neurotransmitter implicated in reinforcing and motivating behaviors as well as movement. MAO-B inhibition is, therefore, associated with enhanced activity of dopamine, as well as with decreased production of hydrogen peroxide, a source of reactive oxygen species [50,51,52]. This, therefore, indicates that the major compound in NVOs has the ability to neutralize the neurotoxicity of venom, which inhibit the release of neurotransmitter from exocytosis of the synaptic vesicle at the presynaptic site or bind to the neurotransmitter receptor at the post-synaptic site. Another compound is Panasinsene (2.06%) which is the major sesquiterpene compound of *Panax ginseng* with medicinal and health benefits in preventing neurodegeneration. The beneficial effects of *P. ginseng* on neurodegenerative diseases have been attributed primarily to the antioxidative and immunomodulatory activities of its ginsenoside components [53,54].

In volatile oil (VO) from the root bark, a total of 49 compounds were identified. Generally, according to a literature search, there has been no research performed on the root bark of *Toona* species. Therefore, the result presented about VO reported for the first time about the chemical constituents of VO in the root bark of *Toona* species. The reason why this study focused on the anti-venom potentials of root bark came up as a result of Traditional Medicine Practitioners (TMPs) in Northern Uganda using it for the treatment of snakebites. Medicinal values of major compounds in VO include; Santalene (8.55%); Santalene sesquiterpene is identified for the first time in a large amount in *C. lansium* from Guangxi Province and revealing the presence of *α*-santalene in *C. emarginata*. The present work exhibited that Eos of *Clausena* has an excellent potential for application in the management of the booklice *Liposcelis bostrychophila* [7,55]. Most health benefits are for Santalol. Santalene, a major component of the sandalwood essential oil, is a typical representative of sesquiterpenes and has important applications in medicine, food, flavors, and other fields, and it has been biosynthesized because of limited supplies. Β-Copaene (8.1%); for the first time, a study reports that copaene is not genotoxic and it increases the antioxidant capacity in human lymphocyte cultures in a study conducted on the cytotoxic, genotoxic/antigenotoxic, and antioxidant/oxidant activity of copaene [56]. In this study, volatile essential oil neutralized cobra venom (SDS-PAGE analysis), and other essential oils also did neutralize the venom of other species. Essential oils have anti-venom potentials, essential oils from *C. bonariensis* and *T. diversifolia* inhibited the coagulant activity of *B. atrox* venom by increasing the clotting time from 100.8 to 264.0 s, respectively, and the oils from *Ambrosia polystachya* and *Baccharis dracunculifolia* caused 100% of inhibition on the fibrinogenolysis induced by *Bothrops moojeni* and *Lachesis muta* venoms [57]. Essential oils of *Artemisia herba-alba* and *A. campestris* (Asteraceae) inhibited inflammation and cytotoxicity induced by *Cerastes cerastes* venoms [58].

This study confirmed that some compounds which were previously identified in the leave (L) and stem (St) *of Toona ciliata* were detected and identified in the roots bark of *Toona ciliata* oils. They include; Sesquiterpenes: Cubebene (L), Ylangene (L), Cubebene (L), β-Elemene (L&St), α-Guaiene (L), and Oxygenated sesquiterpenes: Cubebol (L), Globulol (L&St), epi-Cubenol (L), Muurolol (L) and Cadinol (L&St) [26,27].

The main classes of PLA_2_ inhibitors are the phenolic compounds, which include flavonoids (e.g., Quercetin-3-O-rhamnoside), coumestans and alkaloids, steroids and terpenoids (mono-, di-, and triterpenes), and polyphenols (vegetable tannins) [31,59] Sesquiterpenoids (Ar-tumerone) have anti-venom properties [60].

## 4. Materials and Methods

### 4.1. Ethical Approval

Ethical approval for this study was obtained from Gulu University Research Ethics Committee (GUREC) (No. GUREC-003-20) and the Uganda National Council for Science and Technology (UNCST) (No. SS 5207) for the use of animals and human erythrocytes models. The use of poisonous snakes for milking venom was obtained from the Uganda Wildlife Authority (UWA) (No. COD/96/02).

### 4.2. Milking and Lyophilizing of Venom

Healthy poisonous forest cobras (*Naja melanoleuca*) in the Elaphidae family were identified, selected, and quarantined by a herpetologist from UWA. The selected snakes were kept caged in captivity at Kavumba Wildlife Conservation and Research Centre in Wakiso district, central Uganda. All the standard safety precautions, including protective gear and antisera, carried in case of accidental snakebites, were taken prior to, during, and after handling the snakes. Milking of venom was performed following the method of Theakston and Reid [15] by pressing venom glands but with some modifications. The volume of venom milked per bite was measured and put in labeled sample bottles and kept in cold chain at 4–5 °C. The samples were then freshly transported to the Chemistry Department at Makerere University. The venom obtained was lyophilized to dryness were kept in a refrigerator at 4–8 °C.

### 4.3. Source of Plants Material

The selection of plant material was based on a previous ethnopharmacological survey which showed *T. ciliata* to be one of the most frequently used plants for the treatment of all kinds of poisonous snakebites in Uganda [28]. A voucher specimen of the plant was collected using the WHO standard plant collection guidelines on good agricultural and collection practices for medicinal plants [61] and deposited at the Makerere University herbarium for identification (Voucher No. 50912/ODF 012). After identification, *Toona* M.Roem was also checked with http://www.theplantlist.org/tpl1.1/search?q=Toona+ciliata (accessed on 1 March 2023). Collection and of data was from 5 August to 15 October 2017. The roots of *T. ciliata* were harvested from Namokora sub-county in Kitgum district, northern Uganda, following the same WHO guidelines. The root barks were cleaned and shade dried for 2 weeks. The dried sample was pulverized into fine powder and then kept for extractions.

### 4.4. Extraction of the Plant Material

Extraction of 100 g of powdered root bark of *T. ciliata* was performed twice using warm distilled water (soaked and left for 12 h). The extract was filtered using filter funnel packed with cotton wool then filtrate was filtered using Whatman No. 1 filter paper. The filtrate of the aqueous extracts was freeze-dried and kept in fridge (−8 °C) for phytochemical screening and anti-venom potential analysis. Organic extraction was performed with a solvent system made of Dichloromethane (DCM) and ethylacetate (EA) in a ratio of 1:1 and used for the extraction of 0.5 kg of dry powdered root bark of *T. ciliata*. Filtration was performed using Whatman No. 1 filter paper, and the filtrate was concentrated at reduced pressure at 30–40 °C with a rotary evaporator model RE100-Pro China [62]. The dry sample was kept for phytochemical screening and isolation of fractions [63].

### 4.5. Phytochemicals Screening

The aqueous and organic extracts of root bark of *T. ciliata* were screened for alkaloids, flavonoids, tannins, flavonoids, anthraquinones, coumarins, and terpenoids using standard methods [64].

### 4.6. Anti-Venom Potentials of Toona ciliata

#### 4.6.1. Experimental Animals

Mature healthy Wistar albino rats (*Mou norvegicus albinus*) of either sex, whose weight ranged from 100–130 g, were purchased from College of Veterinary Medicine, Animal Resources and Biosecurity (CoVAB), Makerere University. The rats were acclimatized at room temperature before the experiments and kept in well-sanitized constructed wooden cages at the Department of Veterinary Pharmacy, Clinics, and Comparative Medicine, Makerere University. The rats had free access to standard pellet diet and water.

#### 4.6.2. Lethal Dose (LD_50_) Determination of Venom

A stock solution of the venom was prepared by dissolving 0.01 g of *Naja melanoleuca* venom crystals in 10 mL of normal saline. The LD_50_ of *N. melanoleuca* venom was determined by injecting different dose concentrations of venom (5, 2.5, 1.25, 1, 0.625, 0.5, 0.25, 0.15, and 0.005 µg/g) from stock solution into the tail vein intravenously (i.v.) compared with the intramuscular route (i.m.). Both routes (i.v and i.m) used to help determine which route rats die faster. Groups of rats (*n* = 4) were used, and mortality recorded. Survival time for each group was recorded, and LD_50_ was determined using Probit statistical method [64].

#### 4.6.3. Neutralization of Venom by Aqueous Extract

##### Administration of the Mixture of Venom and Aqueous Plant Extract

In order to find out whether the plants extract neutralized venom, an experiment was run using venom concentration of 1.25 µg/g as control. The solvent, normal saline, used as a control in LD_50_ determination showed no mortality, and it was left out. The stock solution of venom in normal saline was made at a concentration of 1 mg/mL, and that of plant extract was made at 100 mg/mL. A concentration of 1.25 µg/g (from 1 mg/mL) venom dose was mixed with 3.5 mg/g (100 mg/mL) aqueous extract. Similarly, A concentration of 1.25 µg/g (from 1 mg/mL) venom was mixed with 3.5 mg/g (100 mg/mL) aqueous extract. The mixtures of venom and aqueous extract were incubated at 25 °C for 20 min and centrifuged (Centrifuge 5424R, Germany) at 2000 rpm for 10 min. Group I was administered with control venom only. Group II was administered with the resultant mixture intramuscularly and observed for 48 h and days [65,66].

### 4.7. Inhibition of Anticoagulant Activities of Venom

#### 4.7.1. Determination of Minimum Coagulation Dose of Plasma (MCD-P) by Venom

The minimum coagulation dose of plasma (MCD-P) of the venom was determined by first evaluating the coagulation activity of *T. ciliata* extracts as described by the method of Theakston and Reid [15]. Human plasma collected from Nakasero blood bank in Kampala, Uganda, was citrated by adding 4% trisodium citrate as an anticoagulant. The lyophilized venom was used to make a stock solution of 1 mg/mL from which different concentrations from 1–120 µg/mL were diluted with phosphate-buffered saline (PBS) at pH 7.4. Various concentrations of 12.5, 25, 30, 40, 50, and 100 µg/mL of the venom were mixed with 200 µL of the plasma and incubated at 37 °C for 30 min. The plasma was re-calcified to remove anticoagulants with 25 mM CaCl_2_. The PBS alone was used as a control. The setup was observed every 15 s by gentle tilting until a solid thrombus formed, and this was recorded as a recalcification time in minutes. The minimum amount of venom that clots or that allows the formation of a complete thrombus was taken as the minimum coagulation dose of plasma. MCD was calculated by plotting clotting time against venom concentration and reading the level at the 60 s clotting time [15,66].

#### 4.7.2. Determination of Effects of *T. ciliata* Extract on Coagulation of Plasma

To determine if *T. ciliata* extracts have effects on the clotting times of plasma, 100 µL of extracts of concentrations 100, 50, 25, 12.5, 6.25, 1, 0.5, and 0.25 mg/mL were mixed with 200 µL of citrated plasma and incubated in a water-bath at 37 °C. The setup was observed every 5 min for the first 1 h and every 10 min for 2 more h. Coagulability and incoagulability were recorded. After 3 h of observation, 100 µL of 25 mM CaCl_2_ was added to the unclotted plasma samples, and the recalcification time was determined to evaluate the effect on coagulation time.

#### 4.7.3. Neutralization of Anticoagulation Activity of Venom

The minimum coagulation dose of venom determined was doubled (2 MCD-P) and mixed with varied concentrations of plant extracts (6.25, 1, 0.5, 0.25, and 0.125 mg/mL) incubated for 30 min at 37 °C. Then 100 μL of the resultant mixture was added to 200 μL of citrated plasma and re-calcified by the addition of 100 μL of 25 mM CaCl_2_. The clotting times were recorded by gently tilting the test tube containing the sample every 15 s till coagulation took place [15,66].

### 4.8. Inhibition of Phospholipase A2 Activity of Venom

#### 4.8.1. Determination of Minimum Hemolytic Dose of Venom and *T. ciliata* Extract

Evaluation of phospholipase A2 activity of venom by indirect hemolytic activity was assayed as described by Theakston and Reid [15]. A human blood agar gel plate was made containing (300 μL Egg yolk, 0.01 M CaCl_2_ and 1% (*w*/*v*) of Nutrient agar) as follows; 300 μL egg yolk (6 egg yolk/l) solution was added to the normal saline solution, then mixed with 250 μL of 0.01 M CaCl_2_ solution was in a ratio of 1:3 then added to 25 mL of 1% (*w*/*v*) of Nutrient agar (powder) at 50 °C dissolved in PBS. Packed human erythrocytes (300 μL) were washed four times with normal saline solution and then added to the resultant mixture of egg yolk, CaCl_2_ and agar. The mixture was poured into a petri dish and allowed to form a gel (agarose-egg yolk-human erythrocyte gels plate). Then, 6 mm diameter wells were made and filled with 30 μL of venom samples. After 20 h of incubation at 37 °C, the diameters of hemolytic halos were measured. In order to determine the MHD of venom, 30 μL of solutions containing different amounts of venom concentrations (2, 10, 12, 14, 16, 18, and 20 μg/mL) was applied into the wells. Control wells contained 30 μL of PBS, pH 7.4. After 20 h of incubation at 37 °C, the diameters of hemolytic halos were measured. The MHD was determined as the amount of venom that induced a hemolytic halo of 22 mm diameter [20]. The determined MHD was used with different concentrations of the extracts alone to determine the hemolytic properties of *T. ciliate* extract.

#### 4.8.2. Neutralization of Minimum Hemolytic Dose of Venom

The constant MHD of venom was mixed with varied concentrations (6.25, 1.0, 0.5, 0.25 mg/mL) of the aqueous extracts of *T. ciliate* and incubated for 30 min at 37 °C. The aliquots of the mixture, about 15 μL, were added to wells of agarose-egg yolk-human erythrocyte gels plate and then incubated for 20 h at 37 °C. The MHD of the venom only was used as the control. Venom neutralization was calculated as the percentage inhibition that reduces 50% of the diameter of the hemolytic halo compared to that of the control [15,66].

### 4.9. Fractionation of the Organic Extract of T. ciliata

A medium-diameter column was loaded with silica gel (60 particles 0.063–0.2 mm) 70-230 mesh mixed with n-hexane (Hex). Dry organic extracts (Dichloromethane (DCM): Ethylacetate (EA)) (1:1) of *T. ciliata* (30.0 g) were dissolved in acetone and mixed with 30.0 g of silica gel and dried. The sample in silica gel was then loaded, and isolation started with a 100% n-hexane mobile phase. The eluting solvent system started with Hex: EA = 4:1 followed by Hex: EA 3:2, 1:1 till 100% EA and then started EA: methanol 4:1, 3:2, 1:1 till 100% methanol.

The Thin layer chromatography (TLC) Allugram XTra-sheets SIL G/UV254 (20 × 20 cm) to check the homogeneity of fractions. Fractions were combined, concentrated with a digital rotary evaporator, and dried. Three non-volatile oil was isolated with a solvent system of Hex: EA = 4:1 and 3:2.

#### 4.9.1. Hydro-Distillation of Volatile Oils from Root Barks of *T. ciliata*

Air-dried and pulverized root barks (88.8 g) of *T. ciliata* was hydro distilled using a Clevenger-type glass apparatus for 3 h by following the British Pharmacopoeia [67,68]. The distilled oil was 3.0 mL (3.10%) (weight or density in drop (s) not performed, but the density of oils is 0.918 mL/g) obtained were packed in amber bottles and kept in a refrigerator until analysis.

#### 4.9.2. Anti-Venom Activities of Non-Volatile and Volatile Oils of *T. ciliata* by SDS-PAGE

##### SDS-PAGE Gel Electrophoresis

Samples were prepared as Lyophilized venom was dissolved in normal saline at a concentration of 1 mg/mL; both non-volatile and volatile oils were dissolved in DMSO at a concentration of 20 µL/mL. Sodium dodecyl sulfate (SDS) polyacrylamide gel electrophoresis (PAGE) (SDS-PAGE) was carried out using a Mini-Protean IV Electrophoresis Cell device under denaturing conditions using 12% (*w*/*v*) stacking gel solution (8% acrylamide and 0.8% N,N’-bis-methylene acrylamide) and 4% (*w*/*v*) separating gel solution 0.37 M Tris-HCl (pH 8.8) and 0.1% SDS) [68]. The venom and both oils were mixed in a ratio of 1:2 (venom: oils) and incubated for 20 min at room temperature. A protein loading buffer (Protein marker) was added to each of the samples in the tubes and then heated for ten min. The samples were loaded on SDS PAGE gel as follows; Protein marker, Venom only as control, Venom Mixed with NVO, Venom mixed with VO and samples run for 75 min at a constant voltage of 150 v. The gels were then stained with 0.25% commassie brilliant blue overnight. The gels were then destined for 1- and 30-min using the method of Laemmli [68]. Decreased intensity or disappearance of bands as well as the appearance of bands of different molecular weights in the lanes loaded with venom and extracts/oils were used as reliable indicators of activity.

#### 4.9.3. GC-MS/MS Analysis of Non-Volatile (NVO1, NVO2 and NVO3) and Volatile Oils

The non-volatile and volatile oils soluble in n-hexane analytical grade were auto-injected to Gas chromatography-tandem mass spectroscopy (GC-MS/MS) for analysis. Analysis of the oils was performed using GC-MS/MS (model GCMS-TQ 8040, Shimadzu, Japan equipped with autosampler coupled with detectors MS/MS. Capillary column was a DB-5 MS UI fused silica capillary of length 30 m, diameter 0.25 mm, and thickness 0.25 µm with 5% phenylmethylsiloxane stationary phase.

Injection method: injection mode was splitless, sampling time was 2.0 min. Pressure set at 120.0 kpa, and column flow was 1.77 mL/min. Helium was used as the carrier gas at a flow rate of 1 mL/min. The split ratio was 1:0. One microliter of the diluted oils (in hexane) was injected for analysis. n-Alkane of C8 to C30 were run under the same condition of Kovats indices determination as control, the total GC-MS/MS program time was 35.0 min.

Program temperature: The initial oven temperature of the column was set at 60 °C and was heated to the final temperature of 315 °C at a rate of 5 °C/min. The injection temperature was 250 °C. Mass spectroscopy parameter: Ion source temperature was 230 °C, interface temperature was 250 °C, solvent cut time 3 min., scan speed was 1666 at Acq. Mode og Q_3_ scan. Scan time was 60 min with a scanning range of 35–450 amu.

#### 4.9.4. Identification of Chemical Compounds of Both Non-Volatile and Volatile Oils

The constituents of both NVO and VO were identified by GC using retention indices compared with those of the literature. The retention indices were determined in relation to a homologous series of alkanes under the same operating conditions. The components of the oils were identified by matching their spectra and retention indices (Kovats Index) with those of the authentic samples and literature value from the Wiley Library of Mass Spectra database of the GC/MS system and published data [69].

## 5. Conclusions

The compounds in aqueous extract and oils neutralized forest cobra (*N. melanoleuca*) venom. This was evidenced by the aqueous extract of *T. ciliata* neutralizing venom after Pre-incubation of venom with aqueous extract. Since aqueous extract causes blood cloth at some doses, it can be used as a procoagulant drug (a precursor of a natural substance necessary for the coagulation of blood), and it has the anticoagulation activity of venom close to antisera (standard drug). The volatile and non-volatile oils have anti-venom potential; therefore, they can be a good topical application as anti-venom for forest cobra bites. Generally, there has been no research performed on the root bark of *Toona* species. Therefore, aqueous extract and oils of *T. ciliata* showed high potential to be developed as anti-venom and aqueous extract as a coagulant herbal drug. Aroma is one of the most important attributes of food pharmaceuticals and is directly associated with product acceptance by consumers. *T ciliata* has a good yield of essential oil (3.10%) from the root bark. Therefore, this study provides research data for the application of aroma to pharmaceutical products of anti-venom drugs. We recommend future researchers make herbal formulations of medicine with this plant extract for anti-venom and procoagulant. The limitations of this study were the difficulty of hunting wild cobra, which was why we used cage ones, and SDS-PAGE was not showing all the proteins of forest cobra (*N. melanoleuca*) venom clearly.

## Figures and Tables

**Figure 1 molecules-28-03089-f001:**
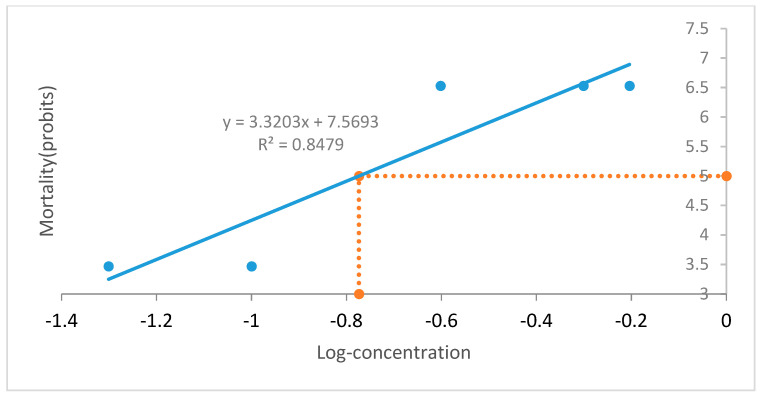
Log_10_ of concentrations of *Naja melanoleuca* venom against the mortality of rats.

**Figure 2 molecules-28-03089-f002:**
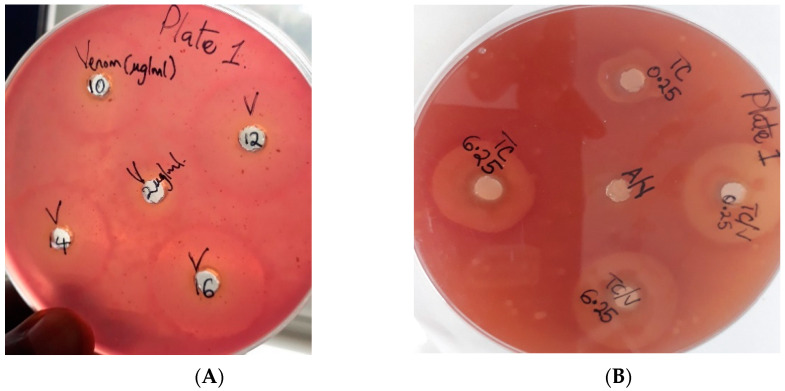
(**A**) Minimum hemolytic dose (MHD), (**B**) Neutralization of PLA_2_ hemolytic activities of venom.

**Figure 3 molecules-28-03089-f003:**
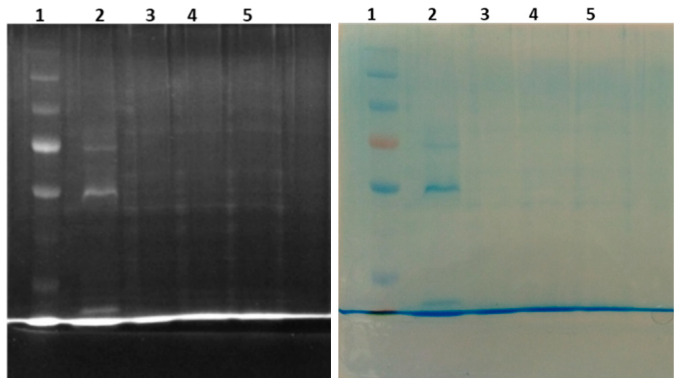
Protein marker, 2—venom, 3—Venom + volatile oil, 4—Venom + Non-volatile oil **B** and 5—Venom + C.

**Figure 4 molecules-28-03089-f004:**
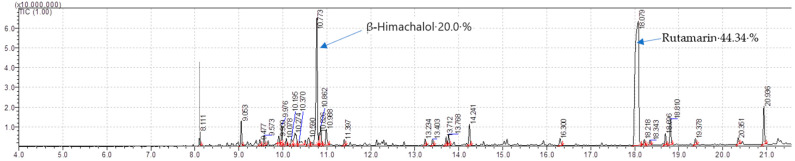
GC-MS/MS Chromatogram of isolated non-volatile oil **A** of root bark of *T. ciliata*.

**Figure 5 molecules-28-03089-f005:**
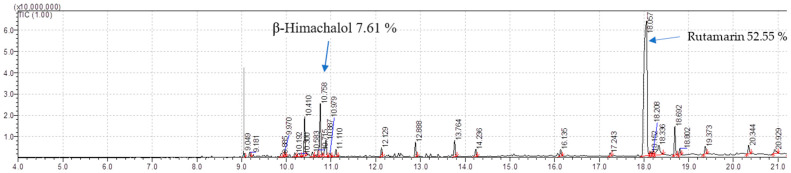
GC-MS/MS Chromatogram of isolated non-volatile oil **B** of root bark of *T. ciliata*.

**Figure 6 molecules-28-03089-f006:**
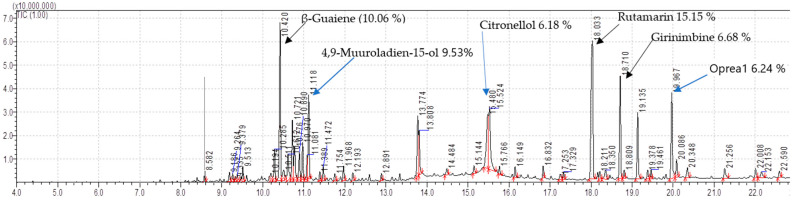
GC-MS/MS Chromatogram of isolated non-volatile oil **C** of root bark of *T. ciliata*.

**Figure 7 molecules-28-03089-f007:**
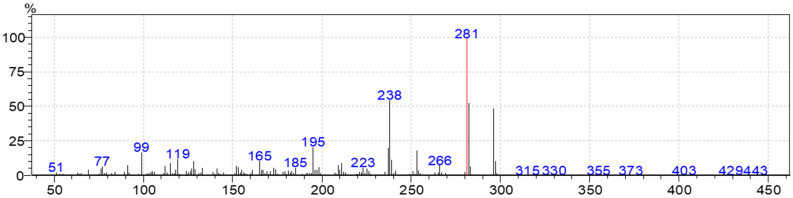
MS/MS chromatograms of Rutamarin in NVOs of *T. ciliata*.

**Figure 8 molecules-28-03089-f008:**
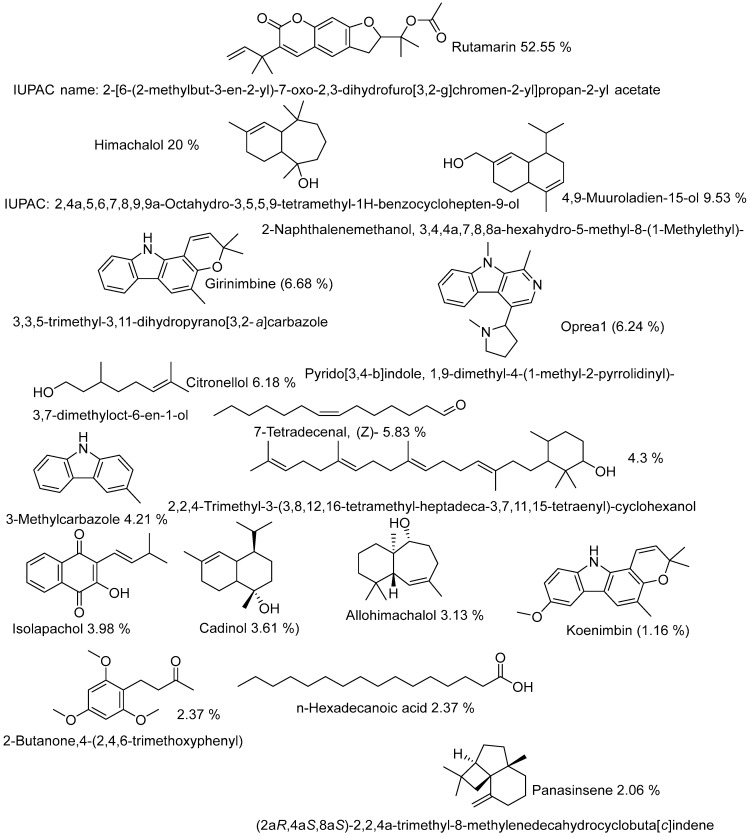
Structure of compounds reported for the first time in root bark non-volatile oils of genus Toona and species *ciliata*.

**Figure 9 molecules-28-03089-f009:**
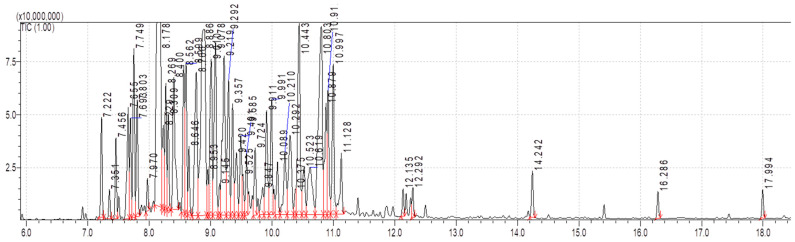
GC-MS/MS Chromatogram of volatile (VO) root bark of *T. ciliata*.

**Figure 10 molecules-28-03089-f010:**
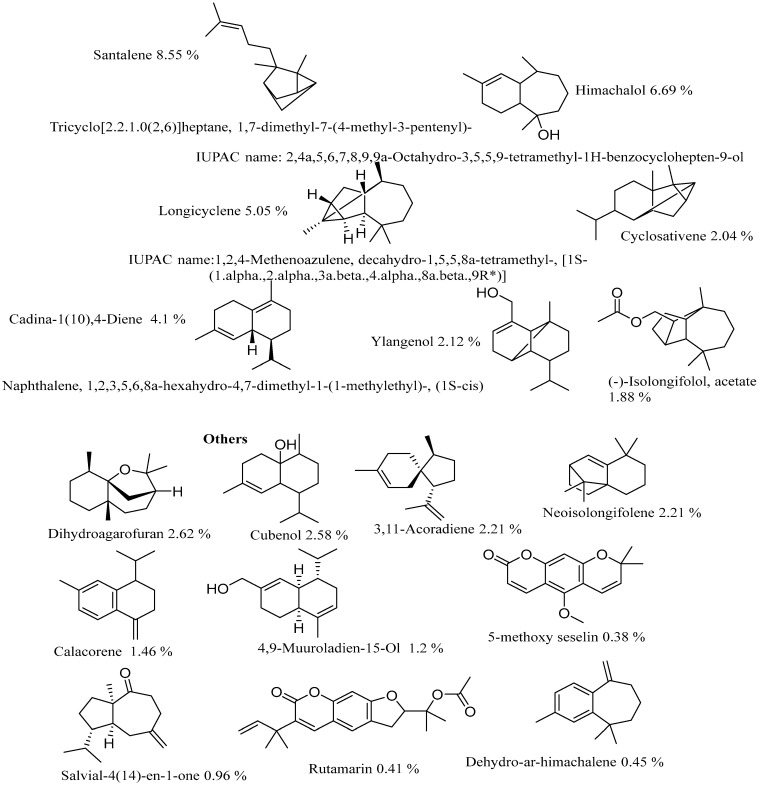
Structures of some of the major compounds reported for the first time in the volatile oil of *T Ciliata*.

**Table 1 molecules-28-03089-t001:** Results of Lethal dose at various concentrations of *N. melanoleuca* venom with their mean survival time.

Group	Concentration (µg/g)	Mean Survival Time (h)	Percentage Death
1	5	0.55	100
2	2.5	1.0	100
3	1.25	2.0	100
4	1.0	2.5	100
5	0.625	3.0	100
6	0.5	4.5	100
7	0.25	7.40	100
8	0.15	12	0
9	0.005	24	0

**Table 2 molecules-28-03089-t002:** Probits for LD_50_ analysis of the venom.

Concentration	Log-Concentration	*n*	*n*	%*n*	*%*n*	Probits
0.625	−0.204119983	4	4	100	93.75	6.53
0.5	−0.301029996	4	4	100	93.75	6.53
0.25	−0.602059991	4	4	100	93.75	6.53
0.1	−1	4	0	0	6.25	3.47
0.05	−1.301029996	4	0	0	6.25	3.47

Key: *n* = number of animals used per dose, *n* = number that died %*n* = percentage death within group, *%*n* = corrected percentage death, percentage death was first corrected for the 0% death [100 (0.25/n] and 100% death [100 (n-0.25/n)].

**Table 3 molecules-28-03089-t003:** Complete neutralization of *N. melanoleuca* venom by aqueous extract of *T. ciliata* root bark.

Experiment		Dose	Group of Mice	Mean Survival Time (h)
Venom	Control	1.25 µg/g	I	2.0
*T. ciliata*	Aqueous Plant extract	3.5 mg/g	II	No death *
*T. ciliata*	Aqueous Plant extract	5.0 mg/g	III	No death *

* The rats were observed for ≥48 h and returned to the cage and lived normal life span.

**Table 4 molecules-28-03089-t004:** Minimum coagulation dose of plasma (MCD-P) by *N. melanoleuca* venom.

	Anticoagulant Activity (MCD-P)
Venom Concentrations (μg/mL)	Mean Clotting time in minutes ± SD
Control (PBS) (no venom)	3.92 ± 0.38
12.5	5.58 ± 0.44
25	9.25 ± 2.68
30	19.58 ± 3.8
40	No clots
50	No clots
100	No clots

**Table 5 molecules-28-03089-t005:** Effect of aqueous extract of *T. ciliata* on coagulation time of plasma.

Extract Concentration in mg/mL	Mean Clotting Time (without Recalcification) (min)	Recalcification Time (min)	
		Exp. 1	Exp. 2
100	22.0	No clots	No clots
50	24.5	No clots	No clots
25	42.5	No clots	No clots
12.5	No clots	2.0	2.17
6.25	No clots	3.5	3.25
1	No clots	4.5	3.75

**Table 6 molecules-28-03089-t006:** Neutralization of coagulation activity of *N. melanoleuca* venom by aqueous extract and SAIMR polyvenom as control.

Aqueous Extract Concentration (mg/mL)	Clotting Time (min)		Mean Clotting Time (min)
	Expt. 1	Expt. 2	
6.25	29.58	15.0	22.29
0.5	15.83	13.72	14.65
0.125	14.83	14.83	14.83
2MCD-P (25 μg/mL)	11.0	11.0	11.0
Antivenin (SAIMR)	9.25	8.08	8.67
Control (PBS)	4.25	5.72	4.99

**Table 7 molecules-28-03089-t007:** The MHD of *N. melanoleuca* venom at various concentrations on human erythrocytes.

Venom Concentration (μg/mL)	2	10	12	14	16	18	20	PBS Control 30 μL
Hemolytic Halos	10	22	24	25	25	25	26	6

**Table 8 molecules-28-03089-t008:** Hemolytic properties of the aqueous extract of *T. ciliata* root bark on human erythrocytes.

Extract Concentrations (mg/mL)	12.5	6.25	1	0.5	0.25	0.125	PBS Control (30 μL)
Hemolytic Halos	18	16	8	6	6	6	6

**Table 9 molecules-28-03089-t009:** Neutralization of PLA_2_ hemolytic activities of venom by aqueous extract.

Extract Concentrations (mg/mL)	6.25	1	0.5	0.25	Control (Venom,10 μg)
Hemolytic halos Mixture	22	22	23	22	24
% Inhibition	8.33	8.33	4.17	8.33	
Hemolytic halos extract only	18	8	6	6	
Control (PBS) only	6	6	6	6	

**Table 10 molecules-28-03089-t010:** Phytocompounds identified in isolated non-volatile oil **A** from the root bark of *T. ciliata* by GC-MS/MS.

No.	R. Time	% Area	Molecular Formula	Name	CAS NO	RI
1	8.111	1.4	C_15_H_24_	Santalene	512-61-8	1463
2	9.053	2.37	C_13_H_18_O_4_	2-Butanone,4-(2,4,6-trimethoxyphenyl)	53581-92-3	1550
3	9.477	0.61	C_15_H_25_I	6-β-Bicyclo [4.3.0]nonane, 5-β-iodomethyl-1-β-isopropenyl-4-α,5-α-dimethyl-,	0-0-0	1589
4	9.573	0.97	C_17_H_28_0_2_	Nerolidyl acetate (Sesquiterpenoid)	2306-78-7	1599
5	9.909	0.8	C_15_H_24_O	Isospathulenol (Sesquiterpenoid)	88395-46-4	1630
6	9.976	1.86	C_15_H_26_O_2_	[5-(Hydroxymethyl)-2,5,8a-trimethyl-1,4,4a,6,7,8-hexahydronaphthalen-1-yl]methanol	1796930-60-3	1636
7	10.078	0.72	C_15_H_24_O	Salvial-4(14)-en-1-one (Sesquiterpenoid)	73809-82-2	1645
8	10.195	1.07	C_17_H_28_0_2_	5-Azulenemethanol, 1,2,3,4,5,6,7,8-octahydro-α,α,3,8-tetramethyl-, acetate, [3S-(3-α,5-α,8-α)]-	134-28-1	1656
9	10.274	2.67	C_15_H_24_O	Humulene epoxide II	19888-34-7	1663
10	10.37	1.28	C_20_H_26_O	Benzene, 1,1′-(oxydi-2,1-ethanediyl)bis [3-ethyl-	55044-9-2	1672
11	10.59	1.29	C_15_H_26_O	τ-Muurolol	19912-62-0	1693
12	10.773	20.0	C_15_H_26_O	β-Himachalol	1891-45-8	1710
13	10.826	1.28	C_15_H_26_O	3-Cyclohexen-1-ol, 1-(1,5-dimethyl-4-hexenyl)-4-methyl-	15352-77-9	1715
14	10.862	2.06	C_15_H_24_	β-Panasinsene	0-0-0	1718
15	11.397	0.58	C_15_H_26_O	(-)-Globulol	489-41-8	1769
16	13.403	0.71	C_14_H_28_O_2_	Dodecanoic acid, 10-methyl-, methyl ester	5129-65-7	1969
17	13.712	1.40	C_16_H_22_O_4_	1,4-Dibutyl benzene-1,4-dicarboxylate or Dibutyl Terephthalate (esther)	1962-75-0	2000
18	13.768	1.1	C_19_H_22_BNO_2_	9H-Carbazole, 9-methyl-	1484-12-4	2006
19	14.24	2.25	C_15_H_26_O	Bicyclo [6.3.0]undec-1(8)-en-3-ol, 2,2,5,5-tetramethyl-	0-0-0	2056
20	16.3	0.9	C_15_H_26_O2	3-Isopropyl-6,7-dimethyltricyclo [4.4.0.0(2,8)]decane-9,10-diol	0-0-0	2286
21	18.079	44.34	C_21_H_24_O_5_	Rutamarin (Furanocoumarins)	13164-5-1	2504
22	18.343	0.58	C_16_H_30_O_4_Si_3_	2,6-Dihydroxybenzoic acid, 3TMS derivative	3782-85-2	2538
23	18.696	1.16	C_18_H_17_NO	Girinimbine (carbazole alkaloid)	23095-44-5	2584
24	20.936	4.3	C_30_H_52_O	β-Himachalol	0-0-0	2894

Key: RT- retention time in the column, RI retention indices relative to n-alkanes C8–C40 on an HP-5 ms column and CAS NO- compound number.

**Table 11 molecules-28-03089-t011:** Phytocompounds identified in isolated non-volatile oil **B** from the root bark of *T. Ciliata* by GC-MS/MS.

No.	RT	% Area	Molecular Formula	Compound Name	CAS NO	RI
1	9.049	0.83	C_13_H_18_O_4_	2-Butanone,4-(2,4,6-trimethoxyphenyl)	53581-92-3	1550
2	9.181	0.76	C_15_H_24_	Santalene	29550-55-8	1562
3	9.885	0.7	C_15_H_26_O	1H-Cycloprop [e]azulen-7-ol,	6750-60-3	1627
4	9.97	0.76	C_15_H_26_O	[5-(Hydroxymethyl)-2,5,8a-trimethyl-1,4,4a,6,7,8-hexahydronaphthalen-1-yl]methanol	1796930-60-3	1635
5	10.3	1.03	C_15_H_26_O	1H-Benzocyclohepten-7-ol, 2,3,4,4a,5,6,7,8-octahydro-1,1,4a,7-tetramethyl-, cis-	6892-80-4	1666
6	10.41	5.93	C_15_H_24_	β-Guaiene	88-84-6	1676
7	10.583	0.87	C_15_H_26_O	τ-Muurolol	19912-62-0	1692
8	10.715	1.24	C_15_H_24_O	1-Aromadendren-10-ol	63181-42-0	1704
9	10.758	7.61	C_15_H_26_O	β-Himachalol	1891-45-8	1709
10	10.887	3.13	C_15_H_26_O	Allohimachalol	19435-77-9	1721
11	10.979	0.78	C_15_H_26_O	β-Acorenol	28400-11-5	1730
12	11.11	1.31	C_15_H_24_O	4,9-Muuroladien-15-ol	135118-51-3	1742
13	12.129	1.16	C_15_H_24_O	cis-Z-α-Bisabolene epoxide	0-0-0	1840
14	12.888	2.92	C_15_H_24_O	6-Isopropenyl-4,8a-dimethyl-1,2,3,5,6,7,8,8a-octahydro-naphthalen-2-ol	0-0-0	1916
15	13.764	2.2	C13H11N	9H-Carbazole, 9-methyl-	1484-12-4	2006
16	14.236	1.07	C_15_H_26_O	Bicyclo [6.3.0]undec-1(8)-en-3-ol, 2,2,5,5-tetramethyl-	0-0-0	2056
17	18.057	52.55	C_21_H_24_O5	Rutamarin	13164-5-1	2501
18	18.152	0.6	C_18_H_21_NO_4_	Oxycodone	76-42-6	2513
19	18.336	3.46	C_16_H_30_O_4_Si_3_	2,6-Dihydroxybenzoic acid, 3TMS derivative	3782-85-2	2537
20	18.692	3.94	C_18_H_17_NO	Girinimbine	23095-44-5	2583
21	20.929	0.94	C_20_H_32_	(E,E,E)-3,7,11,15-Tetramethylhexadeca-1,3,6,10,14-pentaene	77898-97-6	2893

**Table 12 molecules-28-03089-t012:** Phytocompounds identified in isolated non-volatile oil **C** from the root bark of *T. Ciliata* by GC-MS/MS.

No.	R. Time	% AREA	Molecular Formula	Name	CAS NO	RI
1	8.582	0.52	C_12_H_24_	Cyclododecane	294-62-2	1507
2	9.186	0.59	C_15_H_24_	γ-Elemene	29873-99-2	1562
3	9.264	0.45	C_15_H_26_O	(-)-Globulol	489-41-8	1570
4	9.335	0.41	C_17_H_28_O	(-)-Isolongifolol, acetate	0-0-0	1576
5	9.379	0.39	C_11_H_24_O	1-Undecanol	112-42-5	1581
6	9.513	0.77	C_15_H_26_O	Cyclohexanemethanol, 4-ethenyl-α, α,4-trimethyl-3-(1-methylethenyl)-, [1R-(1α,3α,4-β.)]-	639-99-6	1593
7	10.194	0.66	C_15_H_26_O	α-epi-7-epi-5-Eudesmol	446050-56-2	1656
8	10.285	1.86	C_20_H_30_O_2_	Boscartol F	1486443-17-7	1664
9	10.42	10.06	C_15_H_24_	β-Guaiene	88-84-6	1677
10	10.511	1.09	C_15_H_24_O	1H-Cycloprop [e]azulen-7-ol, decahydro-1,1,7-trimethyl-4-methylene-, [1ar-(1a-α,4a-α,7β,7aβ,7bα)]-	6750-60-3	1685
11	10.612	1.97	C_15_H_26_O	1-Naphthalenol, 1,2,3,4,4a,7,8,8a-octahydro-1,6-dimethyl-4-(1-methylethyl)-, [1R-(1-α,4-β,4a-β,8a-β)]-	19435-97-3	1695
12	10.721	3.61	C_15_H_26_O	α-Cadinol	481-34-5	1705
13	10.89	1.75	C_15_H_26_O	(4aR,5R,9aR)-1,1,4a,8-Tetramethyl-2,3,4,4a,5,6,7,9a-octahydro-1H-benzo [7]annulen-5-ol	19435-77-9	1721
14	10.97	2.01	C_15_H_26_O	Cyclohexanemethanol, 4-ethenyl-.alpha.,.alpha.,4-trimethyl-3-(1-methylethenyl)-, [1R-(1-α,3-α,4-β)]-	639-99-6	1729
15	11.118	9.53	C_15_H_24_O	4,9-Muuroladien-15-ol	135118-51-3	1743
16	11.472	0.8	C_15_H_26_O	1H-Cycloprop [e]azulen-7-ol, decahydro-1,1,7-trimethyl-4-methylene-, [1ar-(1a-α,4a-α,7-β,7a-β,7b-α)]-	6750-60-3	1776
17	11.754	0.41	C_15_H_24_O	Ylangenol	41610-69-9	1804
18	12.193	0.49	C_15_H_24_O	((8R,8aS)-8-Isopropyl-5-methyl-3,4,6,7,8,8a-hexahydronaphthalen-2-yl) methanol	135118-52-4	1847
19	13.774	4.21	C_13_H_11_N	3-Methylcarbazole	4630-20-0	2007
20	13.808	2.37	C_16_H_32_O_2_	n-Hexadecanoic acid	57-10-3	2010
21	14.484	0.49	C_10_H_18_O_2_	Bicyclo(3.1.1)heptane-2,3-diol, 2,6,6-trimethyl-	53404-49-2	2082
22	15.144	0.46	C_15_H_20_O_4_	Acetophenone, 4′-hydroxy-2′,6′-dimethoxy-3′-(3-methyl-2-butenyl)-	18780-96-6	2155
23	15.48	6.18	C_13_H_28_OSi	Citronellol, TMS derivative	18419-9-5	2192
24	15.524	5.83	C_14_H_26_O	7-Tetradecenal, (Z)-	65128-96-3	2197
25	15.766	0.42	C_14_H_20_	Bicylo [4.1.0]heptane, 7-bicyclo [4.1.0]hept-7-ylidene-	0-0-0	2225
26	16.149	1.46	C_15_H_24_O	6-Isopropenyl-4,8a-dimethyl-1,2,3,5,6,7,8,8a-octahydro-naphthalen-2-ol	0-0-0	2269
27	16.832	0.64	C_21_H_20_O_2_	2-[1-(2-Acetylphenyl)ethyl]-6-methoxynaphthalene	0-0-0	2350
28	17.253	0.45	C_25_H_41_NO_3_Si	2-Phenanthrenol, 1,2,3,4,4a,4b,5,6,8a,9,10,10a-dodecahydro-4a,7-dimethyl-8-[3-cyano-3-(trimethylsilyloxy)propyl]-, acetate	0-0-0	2401
29	17.329	0.44	C_17_H_13_NO_2_	2-naphthalenol, 1-(2-pyridinyl)-, acetate (ester)	0-0-0	2410
30	18.033	15.15	C_21_H_24_O_5_	Rutamarin	13164-5-1	2498
31	18.35	0.87	C_16_H_30_O_4_Si_3_	2,6-Dihydroxybenzoic acid, 3TMS derivative	3782-85-2	2539
32	18.71	6.68	C_18_H_17_NO	Girinimbine	23095-44-5	2585
33	19.135	3.98	C_15_H_14_O_3_	Isolapachol (1,4-Naphthalenedione, 2-hydroxy-3-(3-methyl-1-butenyl)-)	4042-39-1	2642
34	19.461	0.5	C_14_H_14_N_2_	3-(N-Methylamino)-9-methylcarbazole	5416-98-8	2685
35	19.967	6.24	C_18_H_21_N_3_	Oprea1	54932-49-9	2755
36	20.086	1.16	C_19_H_19_NO_2_	Koenimbin (Carbazoles)	21087-98-9	2772
37	22.008	0.64	C_26_H_52_O_2_	Hexadecanoic acid, dodecyl ester	42232-29-1	3050
38	22.59	0.46	C_26_H_52_O_2_	Hexadecanoic acid, decyl ester	42232-27-9	3127

**Table 13 molecules-28-03089-t013:** Chemical composition of volatile oil (VO) from the root bark of *T. Ciliata* by GC-MS/MS.

No.	R. Time	% Area	Molecular Formula	Name	CAS NO	RI
1	7.222	1.34	C_15_H_24_	Cyclohexene, 4-ethenyl-4-methyl-3-(1-methylethenyl)-1-(1-methylethyl)-, (3R-trans)-	20307-84-0	1378
2	7.351	0.37	C_15_H_24_	α-Cubebene	17699-14-8	1391
4	7.655	2.04	C_15_H_24_	Cyclosativene (1,2,4-Metheno-1H-indene, octahydro-1,7a-dimethyl-5-(1-methylethyl)-, [1S-(1-α,2-α,3a-β,4-α,5-α,7a-β)	22469-52-9	1420
6	7.749	5.05	C_15_H_24_	Longicyclene 1,2,4-Methenoazulene, decahydro-1,5,5,8a-tetramethyl-, [1S-(1-α,2-α,3a-β,4-α,8a-β,9R*)]-	1137-12-8	1429
7	7.803	1.77	C_15_H_24_	(3R,4aS,5R)-4a,5-Dimethyl-3-(prop-1-en-2-yl)-1,2,3,4,4a,5,6,7-octahydronaphthalene	24741-64-8	1434
8	7.97	0.4	C_9_H_11_NO_2_	Benzoic acid, 2-(methylamino)-, methyl ester	85-91-6	1449
9	8.178	8.55	C_15_H_24_	Tricyclo [2.2.1.0(2,6)]heptane, 1,7-dimethyl-7-(4-methyl-3-pentenyl)-, (-)-	512-61-8	1469
10	8.229	1.37	C_15_H_24_	Bicyclo [5.2.0]nonane, 2-methylene-4,8,8-trimethyl-4-vinyl-	242794-76-9	1474
11	8.269	1.82	C_15_H_24_	trans-α-Bergamotene	13474-59-4	1478
12	8.309	1.23	C_15_H_24_	1H-Cyclopenta [1,3]cyclopropa [1,2]benzene, octahydro-7-methyl-3-methylene-4-(1-methylethyl)-, [3aS-(3a-α,3b-β,4-β,7	13744-15-5	1481
13	8.4	2.87	C_15_H_24_	Bicyclo [7.2.0]undec-4-ene, 4,11,11-trimethyl-8-methylene-	13877-93-5	1490
15	8.599	2.21	C_15_H_24_	3,11-Acoradiene (Spiro [4.5]dec-7-ene, 1,8-dimethyl-4-(1-methylethenyl)-, [1S-(1-α,4-β,5-α)]-)	24048-44-0	1508
16	8.646	0.88	C_15_H_24_	(1S,4S,4aS)-1-Isopropyl-4,7-dimethyl-1,2,3,4,4a,5-hexahydronaphthalene	267665-20-3	1513
	8.766	4.68	C_15_H_24_	Copaene (Sesqueterpene)	3856-25-5	1524
17	8.886	8.1	C_15_H_24_	(1R,2S,6S,7S,8S)-8-Isopropyl-1-methyl-3-methylenetricyclo [4.4.0.02,7]decane-rel-	18252-44-3	1535
18	8.953	0.64	C_15_H_24_	Ylangene	14912-44-8	1541
19	9.01	3.24	C_15_H_24_	(1R,4aS,8aR)-1-Isopropyl-4,7-dimethyl-1,2,4a,5,6,8a-hexahydronaphthalene	20085-19-2	1546
20	9.078	4.63	C_15_H_24_	Tricyclo [5.4.0.0(2,8)]undec-9-ene, 2,6,6,9-tetramethyl-, (1R,2S,7R,8R)-	8/2/5989	1553
21	9.145	0.45	C_15_H_20_	α-Dehydro-ar-himachalene	78204-62-3	1559
22	9.219	4.1	C_15_H_24_	Naphthalene, 1,2,3,5,6,8a-hexahydro-4,7-dimethyl-1-(1-methylethyl)-, (1S-cis)-	483-76-1	1566
23	9.292	2.62	C_15_H_26_O	2H-3,9a-Methano-1-benzoxepin, octahydro-2,2,5a,9-tetramethyl-, [3R-(3-α,5a-α,9-α,9a-α)]-	9/2/5956	1572
24	9.357	1.88	C_17_H_28_O_2_	(-)-Isolongifolol, acetate (*Longifolane sesquiterpenoids*)	0-0-0	1578
25	9.42	1.32	C_15_H_22_	4,4-Dimethyl-3-(3-methylbut-3-enylidene)-2-methylenebicyclo [4.1.0]heptane	79718-83-5	1584
26	9.491	1.46	C_15_H_20_	4-Isopropyl-6-methyl-1-methylene-1,2,3,4-tetrahydronaphthalene	50277-34-4	1591
27	9.525	0.52	C_15_H_26_O	Cyclohexanemethanol, 4-ethenyl-α,α,4-trimethyl-3-(1-methylethenyl)-, [1R-(1-α,3-α,4-β)]-	639-99-6	1594
28	9.585	0.96	C_17_H_28_O_2_	Nerolidyl acetate	2306-78-7	1599
29	9.724	1.1	C_15_H_24_	1,5-Cyclodecadiene, 1,5-dimethyl-8-(1-methylethylidene)-, (E,E)-	15423-57-1	1612
30	9.847	0.66	C_15_H_26_O	trans-Sesquisabinene hydrate	145512-84-1	1624
31	9.911	2.12	C_15_H_24_O	Ylangenol	41610-69-9	1630
32	9.991	2.12	C_15_H_26_O_2_	[5-(Hydroxymethyl)-2,5,8a-trimethyl-1,4,4a,6,7,8-hexahydronaphthalen-1-yl]methanol	1796930-60-3	1637
33	10.089	0.96	C_15_H_24_O	Salvial-4(14)-en-1-one	73809-82-2	1646
34	10.21	1.54	C_15_H_24_	isoledene	95910-36-4	1657
35	10.292	2.27	C_15_H_24_	α-Guaiene	12/1/3691	1665
36	10.375	0.44	C_15_H_26_O	(2E,4S,7E)-4-Isopropyl-1,7-dimethylcyclodeca-2,7-dienol	198991-79-6	1673
37	10.443	4.4	C_15_H_24_	β-Guaiene	88-84-6	1679
38	10.523	1.08	C_15_H_26_O	epi-Cubenol	19912-67-5	1687
39	10.619	1.99	C_15_H_26_O	τ-Muurolol	19912-62-0	1695
40	10.803	6.69	C_15_H_26_O	(+)-2-Himachalen-7-ol	1891-45-8	1713
41	10.879	2.21	C_15_H_24_	Neoisolongifolene	26783-22-2	1720
42	10.912	1.9	C_14_H_22_O_2_	Menthol, 1′-(butyn-3-one-1-yl)-, (1S,2S,5R)-	0-0-0	1723
43	10.997	2.58	C_15_H_26_O	Cubenol not Cubebol	21284-22-0	1731
44	11.128	1.2	C_15_H_24_O	4,9-Muuroladien-15-Ol	135118-51-3	1744
45	12.135	0.37	C_14_H_24_O_2_	E,Z-5,7-Dodecadien-1-ol acetate	0-0-0	1841
46	12.292	0.37	C_15_H_28_	1-Pentadecyne	765-13-9	1856
47	14.242	0.7	C_15_H_26_O	Bicyclo [6.3.0]undec-1(8)-en-3-ol, 2,2,5,5-tetramethyl-	0-0-0	2056
48	16.286	0.38	C_15_H_14_O_4_	Xanthoxyletin OR 5-Metoxy seselin	84-99-1	2285
49	17.994	0.41	C_21_H_24_O_5_	Rutamarin	13164-5-1	2493

Key: RT- retention time in the column, RI retention indices relative to n-alkanes C8–C40 on an HP-5 ms column and CAS NO- compound number.

**Table 14 molecules-28-03089-t014:** Summary of class of compounds with percentage abundance.

Class	Non-Volatile Isolated Oils	Volatile/Essential Oil
Hydrocarbon	2.2%	-
2. Sesquiterpene hydrocarbon	6.67%	60.0%
3. oxygenated sesquiterpene (sesquiterpenoids)	48.89%	25.0%
4. Ester	10.0%	2.5%
5. Furanocoumarins (i.e., Rutamarin)	3.33%	2.5%
6. Carbazole alkaloid	7.78%	-
7. Coumarin (Seselin)	-	2.5%
8. Hydrocarbon alcohol	2.2%	5.0%
9. *Longifolane sesquiterpenoids* (i.e., Isolongifolol acetate)		2.5%
10. Alkyl benzene (benzenoids)	1.1%	-
11. Carbazoles (e.g koenimbine)	4.4%	-
12. Harmala alkaloids class	1.1%	-
13. Others (e.g., tetrasiloxane hydrocarbon)	12.33%	-

## Data Availability

Supporting data to this article is publicly available in the Mendeley data repository [70]: Data, V 1, doi:10.17632/n9f4cj8mrn.1.

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
