# Peer review of "In Vitro Anti-Venom Potentials of Aqueous Extract and Oils of Toona ciliata M. Roem against Cobra Venom and Chemical Constituents of Oils"

_molecules, 2023, doi:10.3390/molecules28073089_

Round 1
Reviewer 1 Report
Review comments
The manuscript mainly discusses the anti-venom potentials of aqueous extract and oils of Toona ciliate against cobra venom. Additionally, the authors studied the chemical composition of the oil. This is a good study but some options should be addressed/
1. Are there previous records on the use of herbal products against cobra venom? If include these in the introduction.
2. Give the common name of the plant.
3. Include previous comprehensive studies conducted on this plant.
4. Give the coordination of the collection area.
5. What was the database used for the identification of phytochemicals?
6. Did the authors study AChE activity as the venoms mainly affect the nervous system?
7. Did the authors study only one plant extract dose level or more? The authors should calculate ED50.
8. Compare the effectiveness of the plant extract used with the previously recorded extract.
Reviewer 2 Report
Snakebite envenomation is a neglected tropical disease that is of public health, social and economic importance. It is prevalent in rural communities where there is inadequate health facilities, poor road networks and agrarian activities. Whereas antivenin serum remains the only credited treatment option for ophidian bites, their inaccessibility and unavailability in rural areas has prompted the scientific community to explore natural products as potential sources of antivenom molecules. The study entitled ‘‘In-vitro anti-venom potentials of aqueous extract and oils of Toona ciliata M. Roem against cobra venom and chemical constituents of oils’’ is a study trying to verify the traditional claim of using this species in the management of snakebites in Uganda (https://doi.org/10.1186/s41182-020-00229-4).
Whereas the results are interesting and potentially publishable in MOLECULES, I found several mistakes and omissions in the manuscript that should be corrected before it can be accepted for publication. I request the authors to carefully revise the manuscript to polish it to a standard that is publishable in a journal with global readership.
COMMENTS
1. Title
Could be realigned to read: In vitro anti-venom and phytochemical studies of Toona ciliata root bark extracts and essential oils against black cobra (Naja melanoleuca) venom
2. Abstract
L19-20: Since the species in this study was listed in the ethnobotanical survey done in Uganda (Okot et al. 2020; https://doi.org/10.1186/s41182-020-00229-4), this should be acknowledged and a reason given why this species was selected (among the many species listed) for further investigation.
L23: antiphospholipase A2 (PLA2)?. I suggest revising this to: phospholipase A2 (PLA2) inhibition assays. Also, delete (i.m.) since it was not reused anywhere in the abstract.
L25,28, 33: GC-MS/MS, LD50, T. ciliata and N. melanoleuca needs to be expanded at first use in the manuscript.
L32-33: It will be nice to indicate the major components of the oils first, and thereafter point out those that have been reported for the first time in the species.
37: This line should include the implications of the study, and any recommendations for further research. For example, should isolation and purification be done for any of the compounds identified and tested against the same venom?.
3. Keywords
- antiphospholipase A2 >> phospholipase A2 inhibitor
-Consider adding Neglected Tropical Disease and Snake venom antidote as author-suggested indexing keywords.
4. Introduction
L43: Neglected Tropical Disease (NTD) >> NTD. This was already defined at L41-42.
L48-49: This statement is grossly misleading. There is published information (estimates and retrospective reports) of snakebite incidences in Uganda and Africa. These may be underestimates or even fragmented, but is no different from other areas in the world where snakebites occur. Close to 500,000 snakebite cases are from Africa. In Uganda, 108 cases of snakebites were reported in Gulu Regional Hospital (in 2002) but all the victims did not succumb to the envenomations (Wangoda et al. 2004). Another study indicated a record of 593 snakebite cases in 140 health facilities in a span of six months (April 2018 to June 2018 and then October 2018 to December 2018) in Uganda (Ooms et al., 2020).
Available Literature
UGANDA
Wangoda et al. 2004. Snakebite management: experiences from Gulu Regional Hospital. Uganda. East Cent Afr J Surg. 9:1–5. https://www.ajol.info/index.php/ecajs/article/view/137289
Fact sheet snakebite incidents, response & antivenom supply (Uganda), 2018. https://aidstream.org/files/documents/Fact-Sheet-Uganda-Research-Snakebite-20190128010145.pdf OR Ooms et al. (2020). The Current State of Snakebite Care in Kenya, Uganda, and Zambia: Healthcare Workers' Perspectives and Knowledge, and Health Facilities' Treatment Capacity. The American journal of tropical medicine and hygiene, 104(2), 774–782. https://doi.org/10.4269/ajtmh.20-1078
Omara et al., 2020. Antivenin plants used for treatment of snakebites in Uganda: ethnobotanical reports and pharmacological evidences. https://doi.org/10.1186/s41182-019-0187-0
AFRICA (Other Countries)
Schurer et al. (2022). "At the hospital they do not treat venom from snakebites": A qualitative assessment of health seeking perspectives and experiences among snakebite victims in Rwanda. Toxicon: X, 14, 100100. https://doi.org/10.1016/j.toxcx.2022.100100
Mavoungou et al. (2022) Prevalence and therapeutic management of snakebite cases in the health facilities of the Bouenza department from 2009 to 2021, Republic of Congo. https://www.panafrican-med-journal.com/content/article/42/139/full
L65-66: Since your previous study solely looked at plants used for treatment of snakebites, please revise these lines to: In our previous study, we found that the root bark of T. ciliata (family Meliaceae) is used for treatment of snakebites in Uganda [13]. After this, you may add any other ethnobotanical claims reported by other studies regarding this species.
5. Results
L82: dose doses >> dose.
L103: Unit of LD50 should be indicated.
In Figure 1, the chart title MUST be removed. Axes should be labelled appropriately.
L118-Revise to: Table 3: Complete neutralization of N. melanoleuca venom by aqueous extract of T. ciliate root bark
L180: This should be captured as a Figure.
Section 2.6 should be omitted. It is unnecessary.
L211, 231: Delete ‘‘analysed and’’
6. Discussion
The discussion appears rather weak. The bioactivities need to be correlated with the phytochemicals identified. I did not see anything else to do with the organic extract obtained using DCM: EA (1:1, v/v), why was it extracted?.
7. Materials and Methods
L462: Was this maceration?
8. Author’s Contribution
L621: Please refer to the journal guidelines/manuscript template to enable you rewrite the contributions of the authors.
9. References
Should be realigned to the journal style.
Reviewer 3 Report
I have gone through the manuscript “In-vitro anti-venom potentials of aqueous extract and oils of 2 Toona ciliata M. Roem against cobra venom and chemical 3 constituents of oils”. Overall, it is a well-performed study and an interesting report. I have a few suggestions to improve it further.
1. In page 8, the Authors mention Plate A. It should be numbered as Figure.
2. In Figure 8 and Figure 10, mention in legend/caption, or how you prepared these structures, through chemdraw or copied from some source.
3. Authors should mention the limitations of their study.
4. Conclusion section may be improved by highlighting future perspectives.
5. There are a few typo errors. Language and grammar should be improved in some places.
Round 2
Reviewer 2 Report
The authors have tried to answer most of my concerns. The remaining issue has to do with L53 and is much easier to resolve. Like I had indicated in my previous comments, it is not appropriate to indicate that there is no data regarding snakebites in Africa and Uganda. There are many original articles and reviews on snakebites in Uganda and Africa.
Close to 500,000 snakebite cases are from Africa. In Uganda, 108 cases of snakebites were reported in Gulu Regional Hospital (in 2002) but all the victims did not succumb to the envenomations (Wangoda et al. 2004). Another study indicated a record of 593 snakebite cases in 140 health facilities in a span of six months (April 2018 to June 2018 and then October 2018 to December 2018) in Uganda (Ooms et al., 2020).
Available Literature
UGANDA
Wangoda et al. 2004. Snakebite management: experiences from Gulu Regional Hospital. Uganda. East Cent Afr J Surg. 9:1–5. https://www.ajol.info/index.php/ecajs/article/view/137289
Fact sheet snakebite incidents, response & antivenom supply (Uganda), 2018. https://aidstream.org/files/documents/Fact-Sheet-Uganda-Research-Snakebite-20190128010145.pdf OR Ooms et al. (2020). The Current State of Snakebite Care in Kenya, Uganda, and Zambia: Healthcare Workers' Perspectives and Knowledge, and Health Facilities' Treatment Capacity. The American journal of tropical medicine and hygiene, 104(2), 774–782. https://doi.org/10.4269/ajtmh.20-1078
Omara et al., 2020. Antivenin plants used for treatment of snakebites in Uganda: ethnobotanical reports and pharmacological evidences. https://doi.org/10.1186/s41182-019-0187-0
AFRICA (Other Countries)
Schurer et al. (2022). "At the hospital they do not treat venom from snakebites": A qualitative assessment of health seeking perspectives and experiences among snakebite victims in Rwanda. Toxicon: X, 14, 100100. https://doi.org/10.1016/j.toxcx.2022.100100
Mavoungou et al. (2022) Prevalence and therapeutic management of snakebite cases in the health facilities of the Bouenza department from 2009 to 2021, Republic of Congo. https://www.panafrican-med-journal.com/content/article/42/139/full
